# Experimental Treatments for Oedema in Spinal Cord Injury: A Systematic Review and Meta-Analysis

**DOI:** 10.3390/cells10102682

**Published:** 2021-10-07

**Authors:** Emma Masterman, Zubair Ahmed

**Affiliations:** 1Institute of Inflammation and Ageing, College of Medical and Dental Sciences, University of Birmingham, Edgbaston, Birmingham B15 2TT, UK; emmamasterman@hotmail.com; 2Centre for Trauma Sciences Research, University of Birmingham, Edgbaston, Birmingham B15 2TT, UK; 3Surgical Reconstruction and Microbiology Research Centre, National Institute for Health Research, Queen Elizabeth Hospital, Birmingham B15 2TH, UK

**Keywords:** spinal cord injury, oedema, aquaporin 4, inflammation, functional recovery, TFP

## Abstract

The incidence of spinal cord injury (SCI) is ever-growing, resulting in life-changing neurological deficits which can have devastating long-term impacts on a person’s quality of life. There is an unmet clinical need for a treatment which will prevent progression of the injury, allowing improved axonal regeneration and functional recovery to occur. The initial mechanical insult, followed by a cascade of secondary mechanisms, leads to the exacerbation and remodelling of the lesion site, thus inhibiting neurological recovery. Oedema rapidly accumulates following SCI and contributes to the detrimental pathophysiology and worsens functional outcomes. This study systematically reviewed the current experimental treatments being explored in the field of SCI, which specifically target oedema. Abiding by PRISMA guidelines and strict inclusion criteria, 14 studies were identified and analysed from three online databases (PubMed, Web of Science and EMBASE). As a result, we identified three key modalities which attenuate oedema: selective inhibition of the main water channel protein, aquaporin 4 (AQP4), modulation of inflammation and surgical interventions. Collectively, however, they all result in the downregulation of AQP4, which crucially leads to a reduction in oedema and improved functional outcomes. We concluded that trifluoperazine (TFP), a calmodulin kinase inhibitor which prevents the cell-surface localisation of AQP4, was the most efficacious treatment, significantly eliminating oedema within 7 days of administration. To date, this study is the most concise analysis of current experimental treatments for oedema, exposing its molecular mechanisms and assessing potential therapeutic pathways for future research.

## 1. Introduction

Spinal cord injury (SCI) can result in debilitating conditions that lead to the transient or long-term loss of motor and sensory function as well as impacting people’s physical and social activities and well-being [1,2]. Worldwide, the incidence of SCI has gradually increased in conjunction with the advancement of human activities. Among developed nations, the incidence rate ranges from 13.1 to 163.4 per million people compared to 13.0 to 220.0 per million people in non-developed countries [1,2,3,4]. The aetiology varies among different countries, ethnicities, ages and genders, and is not easily defined due to the variable nature of SCIs and disparity in diagnoses; therefore, the true incidence rate is not captured [2,3,4]. SCIs are often divided into traumatic and non-traumatic origins, resulting from physical trauma to the spinal cord or due to non-external factors such as degenerative, infectious or inflammatory causes. The prevailing cause of SCI continues to be motor vehicle accidents, particularly in developed countries, followed by falls, which dominate in non-developed countries [1,2,3]. This links to the bimodal nature of the age profile of patients with traumatic SCI, peaking at 15 and 29 years and again in older years, >65. The nature of the global aging population has meant that the incidence rate of non-traumatic- and fall-associated SCI is growing within the older generation [2,3,4]. 

Local injury to the spinal cord damages neurons and associated glial cells, disrupts the vasculature and initiates the breakdown of the blood–spinal cord barrier (BSCB) [5,6,7]. Haemorrhaging, oedema and ischaemia are hallmarks of primary injury and lead to a self-propagating progression into secondary injury, which exaggerates and remodels the injury site [8,9]. The extent of the primary injury often determines the severity of the SCI. The breakdown of the BSCB induces haemorrhaging, particularly in the grey matter, which disrupts the micro vasculature, leading to a severe reduction in blood flow to the lesion site and systemic hypotension [10,11]. This, consequently, leads to progressive tissue ischaemia which further contributes to cell death through mechanisms including loss of ATP, oxygen deprivation, excitotoxicity and ionic imbalance [8]. In addition, subsequent reperfusion to the damaged site introduces additional oxygen free radicals and immune cells which contribute to the oxidative stress on endothelial cells, further enhancing BSCB breakdown [8].

Oedema, as previously mentioned, is a key feature in the pathogenesis of SCI and starts to develop within minutes after injury [7]. It further exacerbates the lesion site, progressing it into secondary injury, resulting in a more severe outcomes for the person [12]. It occurs shortly following the primary injury, forming from the epicentre of the lesion site and developing out into the surrounding tissue, to form a fluid-filled cavity around 48 h post-injury [13]. Although the exact biomechanical mechanisms which lead to its formation have yet to be discovered, studies have shown that inhibiting it can greatly improve the prognosis of the person, thereby indicating that the extent of oedema has a subsequent effect on neurological recovery. It is believed that oedema forms as a consequence of dysregulation of water transport regulators within the CNS, predominantly on astrocytes, (cytotoxic oedema) [14] and the mechanical and biological disruption of the BSCB (vasogenic oedema) [13].

Astrocytes are the key regulators of water transport within the CNS. They express aquaporin 4 (AQP4), the main water channel protein in the CNS, within their end-feet and processes as well as the membrane of ependymal cells and play a key role in oedema formation [13,15,16]. Aquaporins are a family of transmembrane protein channels which conduct water transcellularly. To date, 13 members have been identified; however, only AQP1, AQP4 and AQP9 have been identified to play a role in oedema following SCI [13,16]. AQP1 is mostly expressed on the apical membrane of the choroid plexus epithelial cells, ependymal cells and astrocytes within the CNS, and facilitates the production of cerebrospinal fluid (CSF). AQP9, however, is localised to tanycytes, ependymal cells and astrocytes within the CNS, identified to play a role in energy metabolism [17]. AQP4 is the main transporter of water within the spinal cord, and its dysregulation is pivotal to oedema formation. Normally, AQP4 is localised to astrocytes and regulates fast water transport, mediated by tightly controlled osmotic or hydrostatic pressures [13].

To date, no pharmacological interventions have been approved for SCI, and treatment relies heavily on palliative care. However, several potential therapies for oedema have been evaluated in preclinical animal models of SCI. The aim of this study is to systematically review the potential experimental therapies to treat oedema and determine the most potent treatment option for potential translational development.

## 2. Materials and Methods

### 2.1. Review Process

A systematic review of the literature was undertaken, following the guidelines set out by the Preferred Reporting Items for Systematic Reviews and Meta-Analyses (PRISMA) [18]. The search was conducted by two independent reviewers (E.M. and Z.A.) and any discrepancies identified were resolved by consensus.

### 2.2. Literature Search

Published papers which analysed experimental treatments for oedema in SCI were identified following an extensive search of the electronic databases PubMed, Web of Science and Ovid Embase. The following common search string was formulated using the Boolean operators ‘(spinal cord injury) OR (oedema) AND (oedema) AND (rat) NOT (mouse) NOT (reviews)’, in order to encapsulate all the required papers for the study. The search was restricted to studies published from 2011 to 28 January 2021 and in the English language only, to limit the number of results, because most advances in SCI research have taken place within the last 10 years. After exporting to Excel, duplicates were removed, and the studies were screened and shortlisted based upon the on title and abstract, abiding to the outlined inclusion and exclusion criteria. The shortlisted studies were independently reviewed again and approved for full text reading, which produced the final selection of studies for subsequent analyses.

### 2.3. Inclusion and Exclusion Criteria

Studies which were deemed eligible for the systematic review met the following inclusion criteria: (1) studies published in English; (2) published in the last 10 years; (3) articles published in scientific journals; and (4) studies conducted on rats only. We excluded the following: (1) reviews and systematic review articles; (2) studies other than in rats; (3) clinical and in vitro studies; (4) non-English studies; and (5) no full-text articles available. The articles were first assessed based on title and abstract; then, once shortlisted, a full text read was performed by two independent reviewers (E.M. and Z.A.) using an unblinded standardised method to obtain the final selection.

### 2.4. Risk of Publication Bias

The risk of bias for the selected studies was assessed using the SYRCLE’s risk of bias tool for animal studies, adapted from Cochranes’s risk of bias tool [19]. 

### 2.5. Data Extraction

The required data for the study were extracted from the selected articles, obtained using narrative analysis and a meta-analysis, following a full text read. Using pre-designed tables, the basic study characteristics were obtained using a narrative analysis, which qualitatively extracted data from within the publication text. It included the author, the publication date of the study, the location of the experiment, the rat strain used in the animal model, the level of SCI, the type of SCI induced in the experiment, the therapeutic treatment and regime investigated, follow-up timeframe following SCI, and all outcome measures of the study. Secondly, the method the studies used to determine the level of oedema within the spinal cord was also reported. The percentage of spinal cord tissue water content from each experimental group, stated within each investigation, was also noted.

All studies included the percentage of water content in a sham, control and treatment groups. This information, however, was presented in 64% of the publications as a graph and not an exact report of the raw data they obtained, which meant a close estimate of the actual number had to be calculated. The data extracted from the graphs were checked and confirmed by an independent reviewer (Z.A.), and the data used within this systematic review were as accurate as possible. The relative percentage of oedema attenuated by each treatment was then calculated in comparison to the sham and control groups they had described; the standard deviation was also calculated. The processed data were then categorised based upon the follow-up time following the SCI within the experiments for future meta-analysis. Finally, the summary of the other outcomes of the included studies was also extracted. 

### 2.6. Data Synthesis

The data were then synthesised by performing a meta-analysis. The primary purpose of the meta-analysis was to produce a more precise estimate of the effect size of the treatments by combining the effect sizes from the included studies. It was performed on each subgroup analysis, 24 and 72 h post-treatment. We also performed a meta-analysis on the improvement of locomotor function after the various treatments. 

### 2.7. Statistical Analysis

Between-study heterogeneity was determined using the inconsistency index I^2^, which describes the range of possible heterogeneity by the confidence interval. A random-effects models was implemented because heterogeneity was assumed among studies, and it allows for inter-study variability. The effect size was presented as a forest plot, which displays the studies effect size within its corresponding 95% confidence interval. This was carried out using the latest Review Manager 5.4.1 software from Cochrane Informatics & Technology (London, UK), using the random-effects model.

## 3. Results

### 3.1. Study Selection

The systematic search yielded 6615 results across the three databases; PubMed, Web of Science and EMBASE, and after removing duplicates, 6541 remained for preliminary screening of the title and abstract. Abiding by the previously defined inclusion and exclusion criteria, 6525 studies were removed, leaving 16 studies which were deemed eligible for the study and required a full text read. Following the full text screening, two studies were removed because they did not measure spinal cord tissue water content as an outcome measure, therefore 14 studies [3,20,21,22,23,24,25,26,27,28,29,30,31,32] were included within the study for further analysis. The process of literature identification and sorting strategy is displayed by the PRISMA diagram (Figure 1).

One study, Fan et al. [29], was written in Chinese and a full English translation was unavailable. Despite defying the exclusion criteria, following discussion, it was deemed relevant to the study because the data relating to oedema were available in English and was therefore included within the meta-analysis. All of the studies collated investigated experimental treatments to attenuate SCI within their animal models; thus, they underwent further investigation and meta-analysis in order to analyse the current best possible treatment options.

### 3.2. Study Characteristics

Table 1 displays the study characteristics which summarise the general aspects of each study. The location of majority of the publications (78%) were based in China, whereas the remaining studies took place in the United Kingdom, Mexico and the United States. All of the studies, except Cabrera-Aldana et al. [22], utilised Sprague Dawley rats as an experimental animal model, although the gender differed between some, when it was stated. Inducing a spinal cord injury was pinnacle to each investigation, and each performed it at a different spinal cord level, but they all ranged from T6 to T12. Despite all creating an SCI, the method by which this was induced also varied between experiments; 62% of studies used a contusion method, whereas the remining 38% used a compression method.

The therapies, however, differed the most between studies, each investigating a different technique to reduce oedema. They implemented a mixture of pharmaceutical and surgical options each with differing outcome measures. The majority employed pharmacological interventions which involved inhibiting water transport molecules directly, such as TGN-020 or TFP [20,21,30], or indirectly [22,23,24,26,29,33]. Alternatively, some utilised more novel therapeutic options, such as melatonin [27,32], and explored re-purposing therapies with known effects on other ailments. In contrast, other studies researched the effectiveness of surgical techniques, for instance, myelotomy [25], or the insertion of a novel implantable osmotic transport device [31]. Furthermore, the treatment follow-up times also varied, ranging from one hour to six weeks. However, despite this, the majority of studies took measurements at 24 h, 72 h and 7 days, which were consequently grouped for further subgroup analysis (see later). The studies also implemented other outcome measures (Table 1) to analyse other parameters which indicate the treatment effectiveness in reducing SCI. Measuring AQP4 expression was commonly employed, specifically measured by Western blot or immunohistochemistry, and this was often coupled with analysis of the expression of glial fibrillary acidic protein (GFAP). In addition, most studies examined locomotor function of the rat following treatment using the Basso, Beattie, and Bresnahan (BBB) scoring test.

### 3.3. Attenuation of Oedema

#### 3.3.1. Results of Individual Studies

The attenuation of oedema reported by the included studies after treatment ranged from 65% to 85% (Table 2). From this information, it is possible to identify the effectiveness of each treatment option. A large number of the studies reported a decrease in oedema following treatment, which appeared to decrease further over time, returning to sham levels in certain circumstances [20]. Table 2 suggests that Kitchen et al. [20] reported the highest level of attenuation after treatment, indicating TFP as a prominent inhibitor of oedema. Curiously, Cabrera-Aldana et al. [17] demonstrated the largest increase in oedema after treatment with methylpredinoslone (MP) (100% above the sham and control group). However, in contrast, Li et al. [33] reported 72.2% attenuation after MP treatment at 7 days after SCI. Moreover, Liu et al. [27], Ge et al. [28] and Li et al. [32] reported no significant difference in spinal cord water content or AQP4 expression levels at 12 h, but did show a difference at 24 h after treatment. 

#### 3.3.2. Subgroup Analysis

The timescales of analysis of oedema varied significantly between studies; therefore, we sub-grouped studies into three specific time points: 24 h, 72 h and 7 days, to investigate the best possible treatment option and regime (Table 3, Table 4 and Table 5). At 24 h, the percentage of oedema attenuated after treatment across most of the studies was around 50%; however, at 72 h and 7 days, the results were much more variable. For example, the attenuation of oedema varied between 11.1% and 80% at 72 h after treatment (Table 4), whereas only two studies reported an attenuation of oedema at 7 days (Table 5; 72.2% to 108%; Kitchen et al. [20] and Li et al. [32]). 

#### 3.3.3. Functional Outcomes

The studies examined other outcomes in order to explore the effectiveness of their chosen treatment option (Table 6). For example, nearly all studies reported a downregulation of AQP4 following therapeutic interventions compared to the SCI, coupled with a decrease in GFAP too. In addition, some studies performed locomotor function tests in rats, using a variety of methods, including assessment of BBB [20,21,23,24,25,26,33] (Table 7). All of the studies showed improvements in locomotor function, including significant improvements in BBB scores after treatment [21,23,24,25,26]. Studies with significant improvements in BBB also exhibited significant attenuation of AQP4 protein levels after treatment (Table 8). Overall, rats with an improved prognosis tended to be related to a decrease in spinal cord water content and AQP4 downregulation. Additional outcomes were also investigated; however, they were less ubiquitous across the publications and specifically supported their respective study hypotheses.

### 3.4. Meta-Analysis

#### 3.4.1. Oedema

The effect size for oedema after each treatment was compared to the control group within each experiment, in each subgroup analysis. Between-study heterogeneity was assessed by the I^2^ value and an estimate of the between-study variance (T^2^), in a random-effects meta-analysis. At 24 h (Figure 2), the subgroup analysis of the studies had a heterogeneity of 98.98% using the I^2^ statistic, which indicates high heterogeneity between the studies. All treatments, except that used by Cabrera-Aldana et al. [22], produced a statistically significant positive effect, indicating the treatment groups differing significantly from the control groups.

At 24 h (Figure 3), the studies presented a heterogeneity I^2^ score of 91.43%, which also signifies high heterogeneity. The effect size of every study was likewise positively statistically significant, suggesting that the treatment significantly attenuated oedema from the control group.

#### 3.4.2. Locomotor Function Improvements

Out of the seven studies that assessed locomotor function, five studies used BBB, although measurement time-points ranged from 5 to 28 days [22,24,25,26,33]. Despite this, we performed a meta-analysis on the pooled data and calculated an overall mean increase of 5.12 in BBB scores after treatment (95%CI 3.12, 7.11; *p* < 0.00001; I^2^ = 97%) (Figure 4). In addition, only three of the five studies that reported BBB scores also reported AQP4 protein level attenuation, ranging from 53 ± 10% to 73 *±* 5% [22,24,25]. However, these studies demonstrated a negative correlation (−0.73131) between BBB scores and AQP4 protein levels. This suggests that AQP4 levels do not correlate with improvements in locomotor performance.

### 3.5. Risk of Publication Bias

The information provided from the 14 included studies was analysed for the risk of bias (RoB). This process assessed the studies against 10 domains, and from the information presented, they were categorised into yes, yes with insufficient methodology, or not reported (Figure 5). All studies included a description of their primary outcome and supported their thesis with sound methodology, which gave rise to a coherent conclusion. Other than this, the studies varied greatly in the other domains and generally considered a high risk of bias.

Only one study, Hale et al. [31], included an explanation of how they calculated the animal sample size for their study; the rest did not, and only stated the number of animals included. Similarly, 38% of the studies did not report their incomplete data, because the loss of animals was not stated or explained. The majority (62%) of the studies failed to state whether the outcome assessment was randomised; furthermore, only 19% of studies stated that the experimental procedures were blinded to the assessors or animal handlers, but did not provide methodology to support this. Although nearly all the publications detailed the conditions in which the animals were housed, only one study, Zu et al. [24], described housing the animals in a randomised manner. In addition, the allocation of animals to each treatment group was not concealed in 52% of studies. Only 52% of studies implemented randomisation at the correct time, but the methodology or timing of this was not explicitly stated, and judgement was employed as to whether the timing was deemed correct or not, depending on the experimental design. Moreover, 53% stated that they were randomised, and of this, only 10% described how a random sequence generator was used. All of the studies stated the baseline characteristics of the animals utilised within the experiment; however, the methodology of this, its alteration, or relevance to the study were not mentioned. Overall, the bias across studies was deemed high risk, because the data highlight a high percentage of bias across the 10 domains, which may question the validity of the studies.

## 4. Discussion

In this study, we systematically reviewed the published literature within the last 10 years which focused on experimental treatments to alleviate oedema after SCI. The search identified 6541 studies, which were eventually narrowed down to 14 studies after applying our inclusion/exclusion criteria. Following a full text read, the outcome characteristics, including the primary outcome—percentage oedema attenuation—as well as secondary outcomes were extracted and analysed. Studies were grouped into specific follow-up times, 24 h and 72 h, to enable subgroup and meta-analysis. The meta-analysis was conducted to calculate an accurate estimate for the effect size of each treatment. In light of this, the best intervention was the calmodulin (CaM) kinase inhibitor, trifluoperazine (TFP), which blocked the subcellular localisation of AQP4 [20]. TFP significantly attenuated oedema by 108% at 7 days after SCI, reducing oedema to baseline sham uninjured levels. Other treatments failed to completely resolve oedema, although some did significantly attenuate oedema to various degrees.

Our study is novel and impactful because it analysed all of the proposed experimental treatments to reduce oedema after SCI in the last 10 years. As we discovered, there are various ways of reducing injury oedema; however, many of these rely on the suppression of AQP4 levels. In terms of deducing a direct correlation between the level of oedema attenuation and improvements in functional recovery, this was not possible, because none of the studies reported spinal cord water content at the time of functional improvements. However, the reduction in AQP4 levels produced a negative correlation with improvements in functional recovery. This is based only on three studies, and hence must be interpreted with caution because this is not enough evidence to draw definitive conclusions. Moreover, not all drugs targeted AQP4, and not all drugs suppressed AQP4 levels. For example, the most effective treatment for oedema showed that a temporary relocalisation of AQP4 was beneficial in producing significant long-term sensory and functional recovery [20].

Spinal cord oedema is a hallmark of SCI and is a major factor in the propagation of secondary injury. It exacerbates primary injury by increasing the intrathecal pressure, causing further damage through the restriction of blood flow, haemorrhaging and BSCB disruption, and thus inciting further cellular necrosis [5,6,7,8,9,10,11]. Although the exact mechanisms associated with the formation of oedema remain largely unknown, it is strongly associated with an increase in water transport through the AQP4 water channel protein [13,15,16]. Both cytotoxic and vasogenic oedema contribute to the overall excess accumulation of fluid within the spinal cord. For example, ischaemia causes the depletion of ATP, which results in dysfunction of the Na^+^ K^+^ ATPase co-transporter 1 (NKCC1), altering ionic concertation gradients that are normally maintained between the astrocytes and ECM [34]. Oedema occurs within hours of SCI, coincides with ischaemia, and is thought to peak at 7 days after injury [12,20]. Therefore, early treatment with drugs that suppress oedema might be beneficial in improving outcomes after SCI.

Ischaemia also initiates an increase in damage-associated molecular patterns (DAMPs), which elicit an immune response. The surge of immune cells and increase in inflammatory mediators at the lesion site develops a self-propagating amplification of a pro-inflammatory environment, furthering the exacerbation of secondary injury [17]. In addition, the infiltration of immune cells and the production of MMPs and associated ECM molecules leads to the continual disruption of the BSCB [35,36], increasing the permeability of the surrounding capillaries, which are normally tightly regulated. This allows the extravasation of molecules and fluid into the extracellular space of the spinal cord parenchyma, creating vasogenic oedema. Although AQP4 is thought to play a key role in the development of cytotoxic oedema, it is also implicated in vasogenic oedema, but beneficially. AQP4, in later stages of SCI, aids the removal of fluid from the lesion site, clearing oedema, as well as maintaining physiological homeostasis following the chronic stage of SCI [13,37].

Inhibiting oedema soon after SCI has profound benefits to functional recovery. Reducing intrathecal pressure and the development of ischaemia limits the degree of secondary damage and necrosis, facilitating neuroregeneration and the recovery of sensorimotor activity [12]. The downregulation of AQP4 effectively attenuates oedema, but the duality of AQP4 in SCI means that inhibition should be transient [13,20]. Limiting cytotoxic oedema while ensuring AQP4 protein levels are sustained in later stages for elimination of vasogenic oedema and maintaining water homeostasis is important [20]. Although each study investigated a different therapeutic, mechanistically, nearly all reported the modulation of AQP4. Not only was downregulation or the subcellular relocalisation of AQP4 a hallmark of an effective oedema treatment, the degree of oedema attenuation determined the extent of functional recovery [20,21,22,23,24,25,26,27,28,29,30,31,32,33]. To complement these findings, the majority of studies also measured locomotor activity, glial activity using GFAP immunoreactivity and visualised the lesion site using IHC or other imaging techniques in an attempt to understand the anti-oedemic action of the novel treatments as well as the complex pathophysiology behind the development of oedema. The therapies also came under three key modalities: selective AQP4 inhibition; reduced inflammation at the lesion site leading to the downregulation of AQP4; or surgically, with the aim of manually alleviating oedema. To the best of our knowledge, this is the most comprehensive systematic review of experimental treatments proposed in the last 10 years to reduce spinal cord oedema after SCI.

The most successful strategy to treat SCI-induced oedema was to acutely inhibit AQP4 subcellular localisation by a single injection immediately after SCI. Inhibiting CaM kinase, which is required to translocate AQP4 into the cell membrane, maintained normal AQP4 levels, but completely eliminated oedema by 7 days after SCI [20]. This occurred by blocking AQP4 relocalisation to the membrane and stopping its normal function temporarily. TFP, a licensed antipsychotic drug used to treat schizophrenia and short-term severe anxiety, was administered to rats at equivalent doses to humans, and significantly attenuated oedema by 56% at 72 h and completely attenuated oedema by 7 days [20]. The efficacy at 7 days was superior to any other treatment within this systematic review. In addition, acute inhibition of the subcellular localisation of AQP4 with TFP significantly improved electrophysiological, sensory and locomotor function in treated rats, as well as suppressing BSCB breakdown [20]. Acute inhibition of AQP4 relocalisation was key to this success, because the sustained downregulation of AQP4 using a short hairpin RNA against AQP4, although effective at reducing spinal cord water content in the early phase, caused the experiment to be halted as water content reached high levels by 4 weeks after SCI [20]. This reiterates the biphasic nature of AQP4 after SCI; hence, inhibitors to temporarily inhibit AQP4 function should be developed.

Although TFP is reported to work by blocking postsynaptic mesolimbic dopaminergic D1 and D2 receptors in the brain, it is also known to be an antagonist of neuron-specific vesicular protein calcycon, alpha-A1A adrenergic receptor and an inhibitor of calmodulin. We showed that calmodulin kinase (CaM) directly binds to AQP4 and that this binding is inhibited by TFP, suggesting a direct role of TFP in lowering spinal cord oedema. However, kinase inhibitors are often non-specific; hence, there may be other off-target effects of TFP which we have not considered. For example, Midostaurin, a small-molecule inhibitor of multiple kinases such as PKC, caused a significant number of kinomic changes in a cervical SCI model [38]. Nonetheless, we showed that the inhibition of D2 and PKC increased spinal cord water content, demonstrating that the direct and specific inhibition of CaM and PKA is responsible for attenuating SCI-induced oedema after TFP treatment [20].

Other treatments such as TGN-020 are thought to bind to different residues of AQP4 and selectively inhibit AQP4 [39]. However, their ability to modulate oedema is weak, reducing oedema by 44% at best [21,30]. Bumetanide, a diuretic used in renal failure and oedema, blocked NKCC1 and reduced AQP4 expression, correlating with a 33% reduction in oedema after 48 h, and when used in combination with TGN-020, caused attenuation of oedema by nearly 89% at 72 h after SCI [30,39]. Melatonin, a methoxyindole derivative produced from the pineal gland, acts a synchroniser for circadian rhythm, predominantly during the dark phase. It is also known for its antioxidant properties, which can be neuroprotective, predominantly during secondary injury [35]. Melatonin showed limited effects in reducing oedema at 12 h after SCI, peaking to a 50% reduction in oedema after 24 h, and thereafter decreasing to 33% by 42 h. All of these studies reported significant reductions in AQP4 and GFAP expression after treatment, suggesting that the suppression of AQP4 may reduce glial activity after SCI. In addition, melatonin has a plethora of reported effects within the spinal cord, such as reducing oxidative stress, inflammation, NOS, regulating BSCB repair, inhibiting apoptosis, regulating MMPs, as well as affecting other tissues [35].

Along with pharmacological treatments, surgical interventions which mainly decompress the extradural elements in SCI can limit local ischaemia and promote neurological recovery [9,11]. For example, myelotomy removes haemorrhagic and necrotic tissue from the lesion site by opening the dura and pia maters of the spinal cord. Although easily performed in rat models, its application within 24 h of SCI makes it a dangerous procedure which can potentially lead to further complications. The limited reduction in oedema one week following SCI indicates the limited therapeutic benefit of myelotomy [25]. A downregulation of AQP4 and AQP9 expression and an improvement in locomotor function was also detected, indicating that the reduction in oedema was correlated to the reduction in AQP4. Although the study presented beneficial findings, the procedure is too risky for the limited efficacy it delivers, and further studies are required before myelotomy can be considered an option for SCI.

The risk of bias analysis identified significant problems with most of the included studies, ranging from a lack of blinding, inadequate randomisation strategies, and incomplete reporting of excluded animals and prior sample size calculation. The biggest issue was failing to report specific experimental details within the manuscript that relate to risk of bias. Specific guidance on standardised techniques for animal experiments, produced by ARRIVE (Animal Research Reporting of In Vivo Experiments) [40], can help mitigate against a high risk of bias if adhered to, and are recommended for future use in in vivo experiments of this nature. Despite the relatively high tendency of bias, the results obtained are still deemed reliable and enabled significant comparisons and conclusions to be drawn.

### Limitations

One of the main limitations of this study was the variation in time points of the analysis of oedema. However, we were able to group this into subgroups for analysis to within 24 h, 72 h and 7 days after SCI. Another limitation was that there were a variety of locomotor tests used to assess the recovery of function and at time-points ranging from 5 days to 6 weeks. This makes it difficult to compare studies and the potential of reduced oedema and AQP4 levels and their contribution to improvements in locomotor function. Although the majority of animal studies included in this systematic review assessed a thoracic SCI lesion model, 50% of human SCIs affect the cervical region, with C5 being the most common region affected [8]. Key anatomical differences exist between cervical versus thoracic regions of the spinal cord that will affect the severity of injury and the level of oedema experienced after injury. For example, cervical spinal cords are highly vascularised, more permeable blood–spinal cord barrier, exhibit considerable spontaneous recovery and injury interrupts sympathetic innervation to the major immune organs such as the spleen [41]. All these differences will affect lesion development and the attenuation of oedema by the treatments reviewed in this study; hence, further studies are required. Another significant limitation is the potential high risk of bias in some studies, due primarily to a lack of clear reporting. This can easily be mitigated by use of the ARRIVE guidelines in reporting animal experiments.

## 5. Conclusions

This systematic review highlighted three key modalities which can be used to attenuate oedema: inhibiting AQP4, modulating inflammation, and surgical intervention. Although each reduces oedema via different mechanisms, collectively, they downregulate AQP4, which crucially leads to the reduction in oedema and improved functional recovery. Early administration of anti-oedemic therapies, such as TFP, limits spinal cord compression, ischaemia, tissue necrosis and neurodegeneration, and thus improves locomotor function within rat models, reducing spinal cord tissue destruction. It offers a promising outlook for the future of effective SCI treatments, potentially improving patients’ clinical outcomes.

## Figures and Tables

**Figure 1 cells-10-02682-f001:**
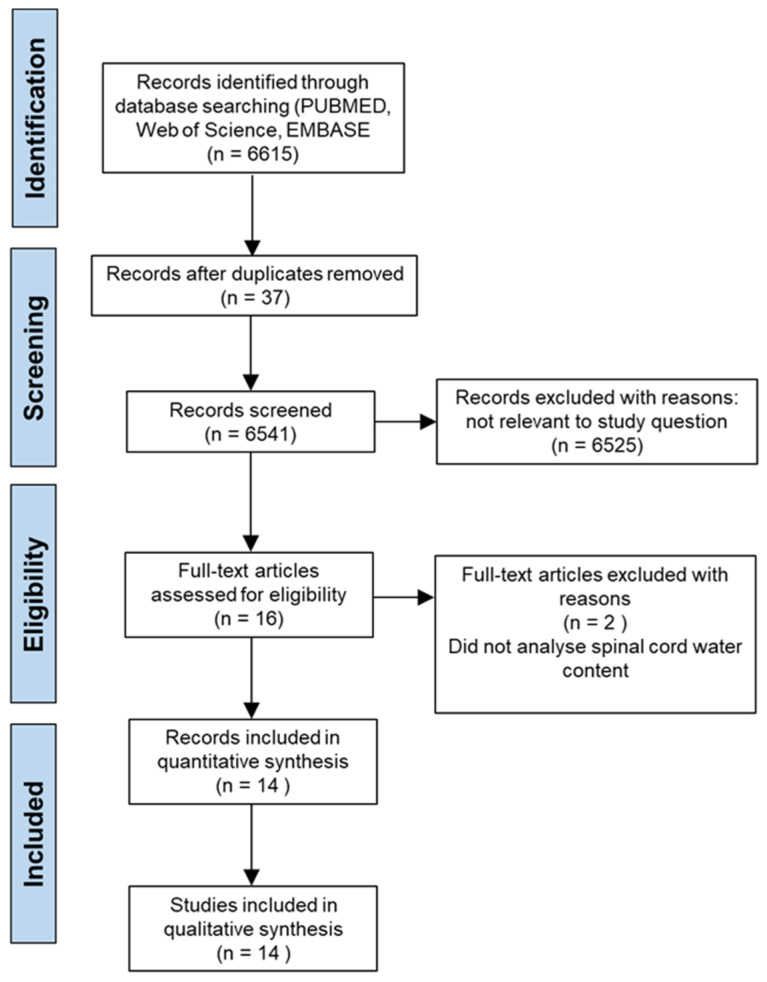
PRISMA flow diagram to demonstrate the screening process for included studies in this systematic review.

**Figure 2 cells-10-02682-f002:**
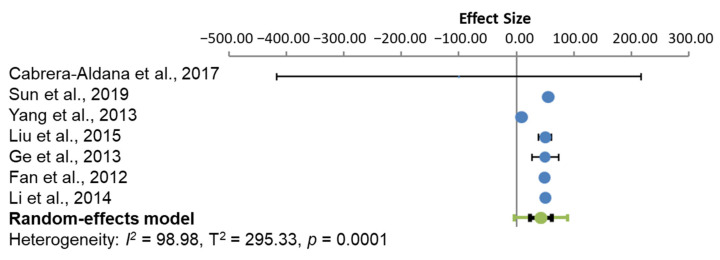
Model of effect size for the subgroup analysis at 24 h after treatment. The circle represents the weighted effect size of the treatment option, and the error bars are the 95% confidence interval for each result. The heterogeneity of the study is represented by the I^2^ statistic and T^2^ is the estimate of variance of the true effect sizes. *p* < 0.001 indicates a statistically significant result.

**Figure 3 cells-10-02682-f003:**
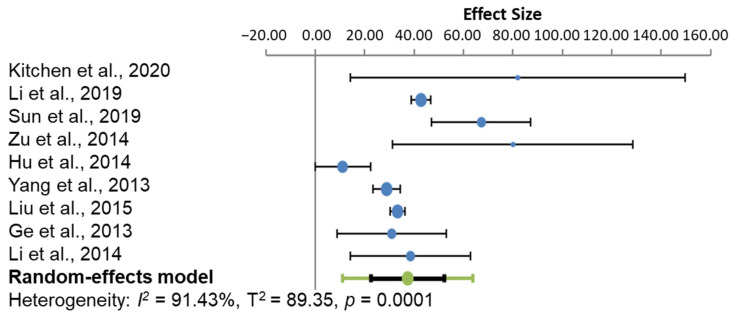
Model of effect size for the subgroup analysis, 72 h from administration of the treatment option. The circles represent the weighted effect size of the treatment option, and the error bars are the 95% confidence intervals for each result. The heterogeneity of the study is represented by the I^2^ statistic, and T^2^ is the estimate of variance of the true effect sizes. *p* < 0.001 indicates a statistically significant result.

**Figure 4 cells-10-02682-f004:**
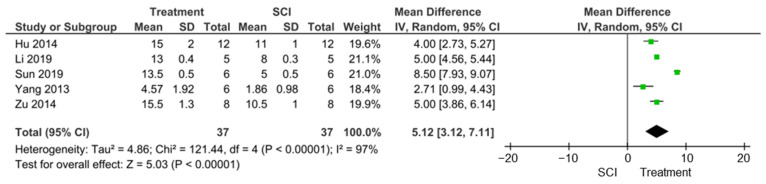
Mean differences in BBB scores after SCI and treatment with appropriate AQP4 inhibitors. The green squares represent the weighted effect size, and the diamond represents the average mean difference of the treatment option. Error bars are the 95% confidence intervals for each result. The heterogeneity of the study is represented by the I^2^ statistic and T^2^ is the estimate of variance of the true effect sizes. *P* < 0.00001 indicates a statistically significant result.

**Figure 5 cells-10-02682-f005:**
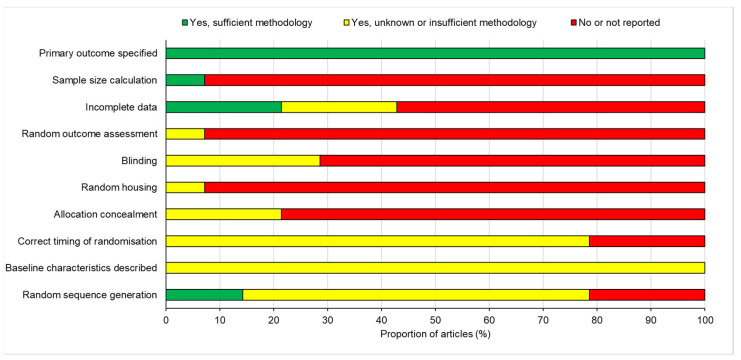
Risk of bias across the 14 included studies.

**Table 1 cells-10-02682-t001:** Characteristics of the included studies.

Study ID	Study Location	Rat Strain	Level of SCI	Type of SCI	Therapy	Follow Up Time after SCI	Outcome Measures
Kitchen et al. [20]	United Kingdom	Sprague Dawley	T8	Dorsal column crush	Trifluoperazine (TFP); Calmodulin kinase inhibitor; protein kinase A inhibitor H89	72 h, 7 days, 28 days and 6 weeks	Water content (oedema); AQP4 IHC; BSCB breakdown; lesion area; electrophysiology; tape sensing and removal; ladder crossing test.
Li et al. [21]	China	Sprague Dawley	T10	Compression	2-(nicotinamide)-1,3,4-thiadiazole (TGN-020),	72 h, 4 weeks	Water content; WB/IF AQP4, GFAP, PCNA, GAP-43 expression; Glial scar formation; neuronal survival; locomotor function.
Cabrera-Aldana et al. [22]	Mexico	M/F Long-Evans	T9	Contusion	Methylprednisolone	24 h	Water content; AQP4, GFAP expression IHC; BSCB breakdown.
Li et al. [33]	China	Sprague Dawley	T6	Contusion	Methylprednisolone	8 h, 24 h, 72 h, 7 days	Water content; HE staining; AQP4 expression WB; Motor nerve function.
Sun et al. [23]	China	F Sprague Dawley	T10	Contusion	Ethyl pyruvate (EP); Glycyrrhizin (GL)	12 h, 24 h, 72 h	Water content; oedema via MRI; AQP4 expression WB/IHC/ELISA; astrocyte expression; TLR4/MyD88 pathway activation; locomotor function.
Zu et al. [24]	China	Sprague Dawley	T8	Contusion	Curcumin	72 h	Locomotor function; HE; water content; AQP4 expression WB/IHC; astrocyte expression.
Hu et al. [25]	China	Sprague Dawley	T10	Contusion	Myelotomy	48 h, 4 days, 6 days	Locomotor function; water content; AQP4, AQP9 expression WB.
Yang et al. [26]	China	M/F Sprague Dawley	T10	Contusion	Hyperbaric oxygen (HBO) therapy	24 h, 48 h, 72 h, 5 days	MMP-2, MMP-9, IL-6 and VEGF expression ELISA/WB; water content; locomotor function.
Liu et al. [27]	China	M Sprague Dawley	T12	Compression	Melatonin	12 h, 24 h, 48 h, 72 h	Water content; AQP4, GFAP expression WB/IHC.
Ge et al. [28]	China	M Sprague Dawley	T12	Compression	Epigallocatechin gallate (EGCG)	12 h, 24 h, 48 h, 72 h	Water content; AQP4, GFAP expression IHC/WB.
Fan et al. [29]	China	Sprague Dawley	NA	NA	Aminoguanidine At 75, 150, 300 mg/kg	0 h, 12 h, 24 h, 48 h	Water content; BSCB permeability; AQP4 expression.
Yan et al. [30]	China	F Sprague Dawley	T12	Contusion	TGN-020 (AQP4 inhibitor); Bumetanide (NKCC1 antagonist)	48 h	AQP4, NKCC1 expression WB/IHC; water content; locomotor activity; LDH activity.
Hale et al. [31]	United States	F Sprague Dawley	T8	Contusion	Implantable osmotic transport device	1 h, 6 h, 12 h, 24 h, 48 h, 72 h, 5 days, 7 days, 14 day, 28 days	Water content.
Li et al. [32]	China	F Sprague Dawley	T12	Compression	Melatonin	12 h, 24 h, 48 h and 72 h	Water content; AQP4, GFAP IHC/WB.

Notes: AQP4, aquaporin-4; BSCB, blood–spinal cord barrier; F, Female; GAP-43, growth-associated protein-43; GFAP, glial fibrillary acidic protein; HE, haematoxylin–eosin staining; IF, immunofluorescence; IHC, immunohistochemistry; LDH, lactate dehydrogenase; M, Male; MyD88, myeloid differentiation primary response 88; NKCC1, Na-K-2Cl cotransporter isoform 1; PCNA, proliferating cell nuclear antigen; T, thoracic; TFP, trifluoperazine; TLR4, Toll-like receptor 4; WB, Western blot.

**Table 2 cells-10-02682-t002:** Level of oedema attenuated. Different treatments within a single study are highlighted in bold.

Study ID	Level of Oedema Reported after Treatment (%)				Level of Oedema Attenuated (%)				
Kitchen et al. [20]	72.8 at 72 h				56.0 at 72 h				
	70.4 at 7 days				108.0 at 7 days				
Li et al. [21]	73.07 at 72 h				42.7 at 72 h				
Cabrera-Aldana et al. [22]	77 at 24 h (G)				−100 at 24 h				
Li et al. [32]	65 (G)				72.2 at 7 days				
Sun et al. [23]	**EP**		**GL**		**EP**			**GL**	
	74 at 12 h		75 at 12 h		55.5 at 12 h			33.3 at 12 h	
	76 at 24 h		77 at 24 h		55.5 at 24 h			44.4 at 24 h	
	74.3 at 72 h (G)		75.5 at 72 h (G)		67.14 at 72 h			50 at 72 h	
Zu et al. [24]	76 at 72 h (G)				80.0 at 72 h				
Hu et al. [25]	76 at 48 h				−9.09 at 48 h				
	72 at 4 days				11.1 at 4 days				
	67 at 6 days (G)				25 at 6 days				
Yang et al. [26]	65.70 at 0 h				165 at 0 h				
	85.67 at 24 h				8.9 at 24 h				
	82.37 at 48 h				18.7 at 48 h				
	78.02 at 72 h				28.8 at 72 h				
	72.97 at 5 days				5.9 at 5 days				
Liu et al. [27]	72.5 at 12 h				16.6 at 12 h				
	72 at 24 h				50.0 at 24 h				
	73 at 48 h				33.3 at 48 h				
	74 at 72 h (G)				33.3 at 72 h				
Ge et al. [28]	73 at 12 h				16.6 at 12 h				
	72 at 24 h				50.0 at 24 h				
	74 at 48 h				30.0 at 48 h				
	75 at 72 h (G)				30.8 at 72 h				
Fan et al. [29]	**0 h**	**12 h**	**24 h**	**48 h**		**0 h**	**12 h**	**24 h**	**48 h**
	78.57	79.82	81.01	80.79	75 mg/kg	16.2	26.8	12.9	37.3
	78.32	78.77	79.81	79.92	150 mg/kg	37.6	62.4	48.5	58.4
	78.11	79.92	81.18	81.57	300 mg/kg	55.5	23.3	8.5	18.4
Yan et al. [30]	**BU** 74				**BU** 33.3				
	**TGN** 73.5				**TGN** 44.4				
	**BU and TGN** 71 at 48 h (G)				**BU and TGN** 88.8 at 72 h				
Hale et al. [31]	72.4 at 3 h				29.03 at 3 h				
Li et al. [32]	72.5 at 12 h				16.6 at 12 h				
	72 at 24 h				50.0 at 24 h				
	73 at 48 h				33.3 at 48 h				
	74 at 7 2 h (G)				38.5 at 72 h				

Notes: G, estimated from graph in manuscript because no raw data were available; EP, ethyl pyruvate; GL, glycyrrhizin; BU, bumetanide; TGN, TGN-020.

**Table 3 cells-10-02682-t003:** Percentage attenuation of oedema: subgroup analysis, 24 h. EP, ethyl pyruvate; GL, glycyrrhizin; SD, standard deviation.

Study ID	Effect Size, i.e., Attenuation of Oedema (%)	N (Animals)
Cabrera-Aldana et al. [22]	−100 ± 99.33	4
Sun et al. [23]	55.5 ± 1.52 (EP); 44.4 ± 2.12 (GL)	6
Yang et al. [26]	8.85 ± 1.64	6
Liu et al. [27]	50.0 ± 4.08	5
Ge et al. [28]	50.0 ± 8.22	5
Fan et al. [29]	12.87 ± 2.74 (75 mg/kg); 48.54 ± 0.44 (150 mg/kg); 8.48 ± 2.15 (300 mg/kg)	5
Li et al. [32]	50.0 ± 2.05	5

**Table 4 cells-10-02682-t004:** Percentage attenuation of oedema: subgroup analysis, 72 h. EP, ethyl pyruvate; GL, glycyrrhizin; SD, standard deviation.

Study ID	Effect Size, i.e., Attenuation of Oedema (%)	N (Animals)
Kitchen et al. [20]	56 ± 13.11	4
Li et al. [21]	42.72 ± 1.69	8
Sun et al. [23]	67.14 ± 7.77 (EP); 50.0 ± 11.64 (GL)	6
Zu et al. [24]	80.0 ± 20.55	8
Hu et al. [25]	11.1 ± 4.04	5
Yang et al. [26]	28.79 ± 2.17	6
Liu et al. [27]	33.3 ± 1.04	5
Ge et al. [28]	30.8 ± 7.96	5
Li et al. [32]	38.46 ± 8.76	5

**Table 5 cells-10-02682-t005:** Percentage attenuation of oedema: subgroup analysis, 7 days. SD, standard deviation.

Study ID	Effect Size, i.e., Attenuation of Oedema (%)	N (Animals)
Kitchen et al. [20]	108.0 ± 10.50	4
Li et al. [33]	72.2 ± 11.25	5

**Table 6 cells-10-02682-t006:** Summary of other outcomes in included studies.

Study ID	Outcomes
Kitchen et al. [20]	Decreased AQP4 IHC in astrocyte end-feet; suppressed BSCB breakdown; reduced lesion area; increased CAP, CAP amplitudes and CAP areas; improved tape sensing and removal times; improved ladder crossing performance.
Li et al. [21]	Decreased AQP4 expression at 72 h; decreased proliferation of astrocytes at 72 h; decreased glial scar formation at 4 weeks; inhibited loss of neurones at 4 weeks; improved locomotor function (BBB scale) at 4 weeks.
Cabrera-Aldana et al. [22]	No improvement in motor outcome following MP; increased impairment of BSCB; increased spinal cord tissue water content following MP; MP decreased AQP4.
Li et al. [33]	Increase in motor neurone function; decrease in oedema volume; reduced haemorrhagic area; decreased AQP4 expression.
Sun et al. [23]	Increase HMGB1 expression following SCI; EP/ GL inhibits HMGB1 expression; improved locomotor function; reduced oedema using MRI; decreased astrocyte expression; reduced AQP4 expression; reduced TLR4/MyD88 pathway activation.
Zu et al. [24]	Improved motor dysfunction; decreased overexpression of AQP4; decrease in astrocyte expression; decrease in activation of JAK/STAT pathway; decreased traumatic manifestations in tissue HE staining.
Hu et al. [25]	Improved locomotor function; decreased AQP4 and AQP9 expression 4 days and 6 days.
Yang et al. [26]	Decreased MMP-2, MMP-9 and IL-6 expression at 48 h, 72 h and 5 days; increased VEGF; Improved locomotor function.
Liu et al. [27]	Decreased AQP4 expression; decreased GFAP expression.
Ge et al. [28]	Decreased AQP4 expression at 4 h and 72 h in IHC; decreased GFAP expression 24 h and 72 h.
Fan et al. [29]	Decreased BSCB permeability; decreased AQP4 levels at 24 h and 48 h
Yan et al. [30]	Decreased tissue destruction; decrease loss of dendrites; decrease in LDH activity; AQP4 and NKCC1 functionally interact with each other.
Hale et al. [31]	Only studied spinal cord water content.
Li et al. [32]	Decreased AQP4 expression; decreased GFAP expression.

Notes: AQP4, aquaporin 4; ASCI, acute spinal cord injury; BSCB, blood–spinal cord barrier; CAP, compound action potentials; EP, ethyl pyruvate; GL, glycyrrhizin; HE, haematoxylin–eosin staining; HMGB1, high-mobility group box-1; IL, interleukin; IHC, immunohistochemistry; LDH, lactate dehydrogenase; MP, methylprednisolone; MMP, matrix metalloproteinases; SCI, spinal cord injury.

**Table 7 cells-10-02682-t007:** Summary of studies which analysed functional recovery, the type of test and its outcome after treatment to reduce oedema.

Study ID	Functional Test	Level of Injury	Type of Injury	Outcome of Functional Test
Kitchen et al. [20]	Tape sensing and removal test and ladder crossing test	T8	DC crush	Sensory and locomotor function returns to sham-treated levels by 3 weeks after treatment.
Li et al. [21]	BBB	T10	Compression	8 ± 0.3 (SCI) vs. 13 ± 0.4 (TGN) at 28 days
Li et al. [33]	Motor nerve function score (MNFS; Tarlov scores)	T6	Contusion	Significantly improved MNFS scores at 3 and 7 days after treatment (81 and 95% of control, respectively)
Sun et al. [33]	BBB and inclined plane (IP) test	T10	Contusion	BBB: 5 ± 0.5 (SCI) vs. 13.5 ± 0.5 (EP); 5 (SCI) and 7 (GL) at 14 days IP: 37 ± 0.05 (SCI) vs. 47.5 ± 0.05 (EP): 37 ± 0.05 (SCI) vs. 41 ± 0.5 (GL) at 14 days
Zu et al. [24]	BBB	T8	Contusion	10.5 ± 1 (SCI) vs. 15.5 ± 1.3 (curcumin) at 14 days
Hu et al. [25]	BBB and incline plane (IP) tests	T10	Contusion	BBB: 11 ± 1 (SCI) vs. 15 ± 2 (myelotomy); IP: 39 ± 1 (SCI) vs. 48 ± 1.7 (myelotomy)
Yang et al. [26]	BBB	T10	Contusion	1.86 ± 0.98 (SCI) vs. 4.57 ± 1.92 (HBO) at 5 days

Notes: BBB, Basso, Beattie and Bresnahan test; DC, dorsal column; EP, ethyl pyruvate; GL, glycyrrhizin; HBO, hyperbaric oxygen; IP, inclined plane test; MNFS, motor nerve function score; T, thoracic. All values estimated from relevant graphs.

**Table 8 cells-10-02682-t008:** Summary of BBB scores before (SCI) and after treatment (SCI + treatment), together with percentage reduction in AQP4 protein by Western blot.

Study ID	BBB Scores			n	Percentage Reduction in AQP4 Protein
	SCI	SCI + Treatment	Effect Size		
Li et al. [22]	8 ± 0.3	13 ± 0.4	5	5	67 ± 6
Sun et al. [33]	5 ± 0.5	13.5 ± 0.5	8.5	6	N/R
Zu et al. [24]	10.5 ± 1	15.5 ± 1.3	5	8	53 ± 10
Hu et al. [25]	11 ± 1	15 ± 2	4	12	73 ± 5
Yang et al. [26]	1.86 ± 0.98 *	4.57 ± 1.92 *	2.71	6	N/R

Notes: *, values given in relevant study. All other values estimated from graphs in respective studies. Values are the mean ± SD. N/R, not reported, AQP4, aquaporin-4.

## Data Availability

All data generated as part of this study are included in the article.

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
