# Peer review of "Experimental Treatments for Oedema in Spinal Cord Injury: A Systematic Review and Meta-Analysis"

_cells, 2021, doi:10.3390/cells10102682_

Round 1
Reviewer 1 Report
Masterman and Ahmed present a very interesting meta-analysis of the experimental treatments for oeadema after SCI. The setting of the analysis, as well as method description of the criteria for selecting the bibliography are very well written and structured. The tables are very resourceful and present important data in a very clear way. The determination of the heterogeneity of the effect sizes for each time point post treatment, is well explained and executed according statistical standars. Additionally ,the inclusion of potential publication bias is very much appreciated. However, and while the meta-analysis is already useful as it is, I believe that it would have a greater impact and would reach broader audiences, if there was an additional analysis of the functional recovery correlated to the attenuation of the oedema, and even to the levels of AQP4, as it is briefly qualitatively mentioned in the subsection 3.3.3. I also believe that, considering the nature and goal of this meta-analysis, these outcomes are not sufficiently discussed, and that would greatly increase the relevance and impact of the meta-analysis.
Minor points:
- In Table 2 notes; Bumetanide after "BU=" is missing
- Line 177 remained instead of remined
- Check punctuation in sentences at lines 401-406
Author Response
Comment: However, and while the meta-analysis is already useful as it is, I believe that it would have a greater impact and would reach broader audiences, if there was an additional analysis of the functional recovery correlated to the attenuation of the oedema, and even to the levels of AQP4, as it is briefly qualitatively mentioned in the subsection 3.3.3. I also believe that, considering the nature and goal of this meta-analysis, these outcomes are not sufficiently discussed, and that would greatly increase the relevance and impact of the meta-analysis.
Author response: We have included a new Table 7, Table 8 and meta-analysis (Figure 4) to discuss these suggestions mentioned by the reviewer. We have compared BBB scores vs the appropriate oedema treatment in studies that reported these values. Table 8 shows improvements in BBB scores after treatment along with % reduction in AQP4 protein, in studies that reported them.
Comment: In Table 2 notes; Bumetanide after "BU=" is missing
Author response: Amended
Comment: Line 177 remained instead of remined.
Author response: Amended
Comment: Check punctuation in sentences at lines 401-406.
Author response: Amended
Reviewer 2 Report
Oedema is one of the immediate hallmarks of traumatic spinal cord injury (SCI), which worsens the damage to the spinal cord. Our current treatments for SCI patients are limited, and oedema presents a promising therapeutic target to prevent further damage to the spinal cord. This study provides a systematic review and meta-analysis of the SCI-specific therapeutics targeting oedema. The authors demonstrate selective inhibition of aquaporin 4 (AQP4), modulation of inflammation and surgical interventions. They conclude that trifluoperazine, a calmodulin kinase inhibitor was a potent treatment to control oedema. While this is a well-written systematic review and meta-analysis, there are a few minor points listed below that need to be addressed by the authors.
- The first paragraph of the introduction is missing citations. Please cite appropriately.
- The authors should discuss the impact and novelty of this study.
- The description for Figure 1 needs to be more informative.
- The timeline following traumatic SCI is quite important, the authors should further emphasize on this.
- The authors need to address level-specific differences in SCI pathogenies. The SCI-induced response is variable between lumbar, thoracic, and cervical injuries.
- Kinase inhibitors are often not specific and have off-target effects. This recent paper on a kinase inhibitor following traumatic SCI explains the role of kinomic alteration in early SCI pathogenesis. The authors should reword their sentences regarding this kinase inhibitor to account for potential off-target effects of trifluoperazine.
Zavvarian, M.-M.; Hong, J.; Khazaei, M.; Chio, J.C.T.; Wang, J.; Badner, A.; Fehlings, M.G. The Protein Kinase Inhibitor Midostaurin Improves Functional Neurological Recovery and Attenuates Inflammatory Changes Following Traumatic Cervical Spinal Cord Injury. Biomolecules 2021, 11, 972, doi:10.3390/biom11070972.
- The authors should proofread the manuscript, there are a few typing mistakes in the text (for example PRISMA is misspelled in the abstract).
Author Response
Comment: The first paragraph of the introduction is missing citations. Please cite appropriately.
Author response: Done.
Comment: The authors should discuss the impact and novelty of this study.
Author response: Done, please see lines 393-404.
Comment: The description for Figure 1 needs to be more informative. Done
Author response: Done.
Comment: The timeline following traumatic SCI is quite important, the authors should further emphasize on this.
Author response: Further emphasised in the discussion.
Comment: The authors need to address level-specific differences in SCI pathogenies. The SCI-induced response is variable between lumbar, thoracic, and cervical injuries.
Author response: Now addressed in lines 538-547.
Comment: Kinase inhibitors are often not specific and have off-target effects. This recent paper on a kinase inhibitor following traumatic SCI explains the role of kinomic alteration in early SCI pathogenesis. The authors should reword their sentences regarding this kinase inhibitor to account for potential off-target effects of trifluoperazine.
Zavvarian, M.-M.; Hong, J.; Khazaei, M.; Chio, J.C.T.; Wang, J.; Badner, A.; Fehlings, M.G. The Protein Kinase Inhibitor Midostaurin Improves Functional Neurological Recovery and Attenuates Inflammatory Changes Following Traumatic Cervical Spinal Cord Injury. Biomolecules 2021, 11, 972, doi:10.3390/biom11070972.
Author response: Addressed in a new paragraph, Lines 472-483.
Comment: The authors should proofread the manuscript, there are a few typing mistakes in the text (for example PRISMA is misspelled in the abstract).
Author response: Done.